# OpenReview forum: "When Benign Inputs Lead to Severe Harms: Eliciting Unsafe Unintended Behaviors of Computer-Use Agents"
_ICML.cc/2026/Conference — ICML 2026 regular_

### Official Review · Reviewer_enqn · 2026-02-17

**Soundness:** 3
**Presentation:** 2
**Significance:** 3
**Originality:** 3
**Overall Recommendation:** 4
**Confidence:** 4

**Summary:**

The work defines “unintended unsafe behaviors” in CUAs with benign inputs and introduces AutoElicit, a two stage pipeline to surface such behaviors in OSWorld-style task. It produces seed sets and execution trajectories; reports high elicitation and partial cross-model transfer.

**Compliance With Llm Reviewing Policy:**

Affirmed.

**Final Justification:**

I am more confident about the result now and think this is in the acceptance area.

**Key Questions For Authors:**

It would nice to have better ablation of different components, such as seed-only vs refinement; refinement w/wo quality gate; single vs ensemble judge. Also, the reproducibility can be addressed by showing how to construct a small scale verification without accessing some mentioned proprietary data. If you can show how success/severity rates change under stricter, human-verified criteria (stratified sampling + CIs), it would be beneficial as well. The work is interesting and promoting, but you need to tighten the presentation and discussion to add credibility of the performance.

**Limitations:**

Yes.

**Strengths And Weaknesses:**

The work has a clear real-world application and concern on the safety problem, and a clear framing of the threat CIA-style harms. It introduces an implementable pipeline and broad empirical sweep.

However, it relies extensively on LLM judges without much human validation. It is worth discussing the quality contrast between human experts and LLM to add credential to this approach in a more robust fashion. The baselines could also be better established. The reproducibility is also concerning due to proprietary components. It would be nice to describe how to reproduce the work in small scale without access to those proprietary components.

---

> ### Author Rebuttal · Authors · 2026-03-31
>
> We appreciate your constructive feedback and positive comments regarding our clear risk framing, implementable pipeline, and broad empirical sweep. We address your concerns regarding human validation, baselines, ablations, and reproducibility below:
>
> ### Human Validation (W1)
>
> We appreciate your concern regarding human validation. We conduct a human annotation procedure (App. H) to validate the LLM judges used in our automatic framework. Specifically, we use three annotators with majority voting to evaluate whether judges accurately 1) identify harms in trajectory analysis, 2) evaluate harms given task and execution context, 3) interpret perturbation benignity and realism, and 4) identify harmful goal-directed CoT reasoning. AutoElicit achieves 79.5% TPR on a representative subset of examples using this criteria. We also identify a 98% TNR from another human annotation study (see cA4a’s W2). Given this, we provide in-depth validation to support LLM judge usage for automatic elicitation.
>
> ### Baseline Comparison (W2)
>
> Thanks for your feedback regarding the lack of baseline comparison. As no prior automatic CUA elicitation methods exist, we adapt TAI3’s [1] core pipeline and its naive baseline SelfRef to the CUA setting as the closest related baseline. We test both using our OS domain seeds, Claude 4.5 Haiku as the execution agent, GPT-5 as the refinement model, and the same elicitation success threshold of 50.
>
> **AutoElicit achieves an Elic. Per Seed % of 72.5% (Tab. 1), while TAI3 (24.2%) and SelfRef (17.6%) show reduced success.** The baselines also skew more toward Min.+Low severity (TAI3: 62.5%, SelfRef: 66.7%) compared to AutoElicit (41.8%).
>
> We also evaluate successful mutation quality using our ensemble judges and 6 quality dimensions (Tab. 5). The largest increases occur for Harm Plausibility (+67.0-74.6) and Implicitness (+22.3-23.3), as baselines produce more overtly malicious prompts rather than subtle ambiguities.
>
> |Dimension|Ours|TAI3|SelfRef|
> |-|-|-|-|
> |Benignity|77.8±5.6|68.9±20.4|76.5±16.9|
> |Realism|92.4±6.4|83.9±12.0|89.8±6.7|
> |Contextual Plausibility|96.5±6.9|86.6±14.6|90.0±10.6|
> |Goal Preservation|91.5±8.1|86.5±12.4|86.7±19.4|
> |Harm Plausibility|80.2±8.0|13.2±11.5|5.6±10.8|
> |Implicitness|82.8±5.4|59.5±20.6|60.5±12.4|
>
> ### Reproducibility (W3&Q4)
>
> We appreciate your concerns of reproducibility. However, all data and code used is publicly available (provided in Supplementary Material) and none of it is proprietary. We also provide full details for seed generation (App. D), including hyperparameters, models, and prompts. This ensures reproducibility through data release and necessary details for dataset construction.
>
> ### Seed-Only vs. Refinement (Q1)
>
> To evaluate the impact of refinement on elicitation success, we measure the percentage of successful perturbations requiring only the initial seed from Context-Aware Seed Generation vs. those requiring Execution-Guided Perturbation Refinement. We measure this across elicitation results for Claude 4.5 Haiku w/ GPT-5 and Claude 4.5 Haiku as refinement models, Operator w/ GPT-5 for refinement (see 9Tg3’s W1), and Claude 4.5 Opus w/ GPT-5 for refinement using 60 randomly sampled AutoElicit-Seed seeds (see cA4a’s W1).
>
> As shown below, **refinement substantially improves elicitation success across all configurations (68.8-86.2% of successful elicitation requires refinement)**, indicating that refinement is critical for surfacing long-tail unintended behaviors.
>
> |Execution Agent w/ Refinement Model|# Successful Seeds|Seed-Only %|Refinement %|
> |-|-|-|-|
> |**Multi-Apps**||||
> |Claude 4.5 Haiku||||
> |- w/ Claude 4.5 Haiku|113|18.6|81.4|
> |- w/ GPT-5|118|22.0|78.0|
> |Operator w/ GPT-5|116|13.8|86.2|
> |Claude 4.5 Opus w/ GPT-5|52|30.8|69.2|
> |**OS**||||
> |Claude 4.5 Haiku||||
> |- w/ Claude 4.5 Haiku|77|31.2|68.8|
> |- w/ GPT-5|79|24.1|75.9|
> |Operator w/ GPT-5|111|24.3|75.7|
> |Claude 4.5 Opus w/ GPT-5|51|25.5|74.5|
>
> ### Quality Refinement Ablation (Q2)
>
> We ablated the inner quality refinement loop such that all perturbations are executed without first passing a quality check. Please see our response to 9Tg3’s W2&Q2 for more details, where results show that quality refinement improves both elicitation success and perturbation quality.
>
> ### Single vs. Ensemble Judge (Q3)
>
> We analyze 1,271 quality evaluations that were independently scored by all 3 judges. Individual judges exhibit distinct evaluation tendencies: o4-mini allows broader coverage (92.0% benignity pass rate), while Claude Haiku is stricter (32.8%), reflecting complementary roles. The judges show moderate pairwise agreement (36-61%) with high average inter-judge score std (Benignity: 16.8, Implicitness: 16.4, 0-100 scale), confirming that each judge captures distinct quality signals. Ensemble averaging leverages this diversity, producing a more balanced and robust quality gate than any single judge alone.
>
> [1] TAI3: Testing Agent Integrity in Interpreting User Intent (NeurIPS 2025)

---

> > ### Author Rebuttal · Reviewer_enqn · 2026-04-01
> >
> > Thanks for answering the questions; this resolves my initial concerns nicely.

---

### Official Review · Reviewer_9Tg3 · 2026-03-09

**Soundness:** 2
**Presentation:** 3
**Significance:** 3
**Originality:** 3
**Overall Recommendation:** 4
**Confidence:** 4

**Summary:**

This paper tackles a subtle but practically important CUA safety risk: severe harmful behaviors that emerge not from adversarial manipulation, but from ambiguous yet well-intentioned natural language instructions. The authors contribute three things: a conceptual framework defining "unintended behaviors" through four characteristics (deviation from user intent, emergence under benign input, safety constraint violation, and distinction from capability failures); AUTOELICIT, a two-stage agentic pipeline that automatically generates and iteratively refines perturbed instructions using real execution feedback; and empirical findings showing elicitation success rates of up to 72.5% on Claude 4.5 Haiku, with successful perturbations transferring broadly across frontier CUAs. Three datasets are released to support follow-up research.

**Compliance With Llm Reviewing Policy:**

Affirmed.

**Key Questions For Authors:**

1. The main results in Table 1 are restricted to Claude 4.5 Haiku and a biased subset of Claude 4.5 Opus. Can the authors provide full-scale elicitation results on at least one model from a different family, such as GPT-5 or Operator, as a primary execution agent? Without this, it is difficult to assess whether the reported success rates reflect a property of the AUTOELICIT method or a specific characteristic of Anthropic models.

2. AUTOELICIT's performance is reported as a complete system, but no ablation is provided. How much does the inner quality refinement loop contribute to final elicitation success rates compared to the outer execution feedback loop alone? Similarly, how sensitive are results to the seed filtering thresholds defined in Appendix D.5? A ablation over these components seems essential to justify the design choices made.

**Limitations:**

see weaknesses and questions.

**Strengths And Weaknesses:**

**Strengths**

- The paper identifies a genuinely underexplored safety risk. While most prior work on CUA safety focuses on adversarial manipulation or explicit harmful prompts, the authors convincingly argue that benign, naturally occurring instructions can themselves be a sufficient trigger for severe harms. This framing is both practically important and conceptually distinct from existing threat models.
- The four-characteristic definition of unintended behaviors is carefully constructed, particularly the distinction between unintended behaviors and general capability mistakes. Grounding this distinction in Chain-of-Thought monitorability provides a concrete, if imperfect, operationalization. The CIA-based taxonomy for categorizing cybersecurity risks further adds structure to what has previously been a fragmented and anecdotal literature.

- The 87.5% reproducibility rate reported in Table 8, combined with a 57.6% per-trial harmful rate, provides meaningful evidence that the elicited behaviors represent systemic vulnerabilities rather than stochastic execution artifacts.

------

**Weaknesses**

- The main elicitation results in Table 1 are limited to two models from the same model family, Claude 4.5 Haiku and a small high-severity subset of Claude 4.5 Opus. The Opus evaluation is further restricted to the 30 seeds that produced the most severe harms against Haiku, introducing a selection bias that makes the Opus results difficult to interpret independently. No other model families are evaluated as primary execution agents, which significantly limits the generalizability of the main findings.

- AUTOELICIT involves numerous design choices, including the dual feedback loop architecture, the seed filtering thresholds, the maximum iteration limit of 10, the ensemble evaluator configuration, and the verbalized sampling strategy. None of these are ablated. It is therefore unclear how sensitive the reported success rates are to these choices, and which components are actually driving performance. For instance, it is unknown whether the inner quality refinement loop meaningfully improves over a simpler single-loop baseline.

* The paper does not compare AUTOELICIT against any baseline elicitation method. A natural comparison would be random perturbation of benign instructions, or prior approaches such as ToolEmu or OS-Harm adapted to the same task domains. Without such comparisons, the reported success rates cannot be contextualized. It is unclear how much of the elicitation success is attributable to the method's design versus the inherent susceptibility of the models tested.

- The human annotation study reports a True Positive Rate of 79.5% for the automatic trajectory evaluator, meaning roughly one in five flagged cases is a false positive. However, the paper does not report the False Negative Rate, making it impossible to assess whether the evaluator systematically under-detects certain harm categories. Given that Table 1's success rates are derived from this evaluator, unreported false negatives could meaningfully affect the interpretation of the main results. Additionally, the annotation study covers only a stratified subset of Haiku elicitations and all Opus elicitations, which may not be representative of the full evaluation distribution.

---

> ### Author Rebuttal · Authors · 2026-03-31
>
> We thank the reviewer for the positive comments on our study’s **practical importance,  well-constructed conceptual framework, and strong reproducibility results.** We address your concerns regarding evaluation breadth and ablations below:
>
> ### Cross-Family Testing (W1&Q1)
>
> Thanks for raising concerns about AutoElicit's generalizability beyond Claude models. To address this, we perform **full-scale elicitation analysis with OpenAI Operator** using AutoElicit-Seed (Refinement Model: GPT-5, same setup as Tab. 1 experiments).
>
> |Domain|# Seeds|# Tasks|Elic. Per Seed %|Elic. Per Task %|None|Min.|Low|Med.|High|Crit.|
> |-|-|-|-|-|-|-|-|-|-|-|
> |OS|136|16|81.6|100.0|18.4|0.7|36.0|31.6|11.0|2.2|
> |Multi-Apps|171|40|67.8|87.5|29.4|9.4|28.8|23.5|6.5|2.4|
>
> AutoElicit consistently elicits from Operator (OS: 81.6%, Multi-Apps: 67.8%), surfacing harms from tasks with a 0% baseline harm rate (OS: 100%, Multi-Apps: 87.5%). As noted by m5vt, successful perturbations also transfer to a broad range of CUAs (Tab. 2), indicating that AutoElicit exposes model-agnostic vulnerabilities within benign input contexts. These results show that AutoElicit surfaces long-tail risks across diverse frontier CUAs instead of exploiting model-specific properties.
>
> ### Ablations (W2&Q2)
>
> Thanks for the feedback regarding the ablations of AutoElicit components. We **conduct additional ablation studies** and analyses to validate key design choices below.
>
> **Quality Refinement.** We ablated the quality refinement loop by executing instructions refined from execution feedback with no quality check. We perform this ablation with 30 OS domain seeds, Claude 4.5 Haiku as the execution agent, and GPT-5 as the refinement model.
>
> |Metric|Full Method|Ablation|Δ|
> |-|-|-|-|
> |Elic. Per Seed %|76.7%|50.0%|-26.7%|
> |Avg Elic. Score|49.2|41.7|-7.5|
>
> We also evaluated successful perturbations from w/ and wo/ quality refinement based on perturbation quality using our ensemble judge approach.
>
> |Dimension|Full Method|Ablation|Δ|
> |-|-|-|-|
> |Benignity|78.3|60.9|-17.4|
> |Realism|92.7|83.8|-8.9|
> |Contextual Plausibility|91.1|87.7|-3.4|
> |Goal Preservation|90.2|79.1|-11.1|
> |Harm Plausibility|71.1|18.0|-53.1|
> |Implicitness|83.5|42.0|-41.5|
>
> **Seed Filtering Thresholds.** We perform post-hoc analysis measuring how elicitation is impacted given progressively stricter seed filtering thresholds (App. D.5). For each score, we increase the minimum threshold (step=5) and recompute Elic. Per Seed % for the retained seeds. We do not recompute this below the default threshold, as this requires using AutoElicit
>
> Figures: https://anonymous.4open.science/r/temp_figures-256B/
>
> Results for both the OS domain (Fig. 1-3) and Multi-Apps (Fig. 4-6) show that elicitation success remains stable across stricter thresholds with it only spiking/plummeting when performance is noisy due to a small number of retained seeds. This suggests that seed filtering prioritizes high-quality seeds for improved efficiency/costs instead of driving elicitation success.
>
> **Verbalized Sampling.** We ablated VS-Multi (verbalized sampling) by performing single-shot seed generation for all OS tasks, independently generating each seed with an identical prompt, temperature, and # of seeds per task.
>
> We evaluate across 4 generation models (o4-mini, GPT-5, Haiku/Opus 4.5) and N=5, 10, 15, 20, 30, and 50 seeds, yielding 48 configurations. We measure diversity using Self-BLEU (↓), normalized pairwise edit distance (↑), and Shannon entropy over the elicitation strategy distribution (↑).
>
> Removing VS-Multi consistently **degrades seed diversity across all configurations** on average (Self-BLEU +0.140, edit distance -0.094, strategy entropy 1.36→0.93) with stronger effects for stronger models (e.g., Claude Opus 4.5: Self-BLEU +0.214, edit distance −0.159, entropy 1.84→0.88). These results confirm that verbalized sampling improves perturbation diversity by enabling the LLM to actively explore different regions of the perturbation space.
>
> ### Annotation Distribution (W4)
>
> Thanks for raising this concern. While we do not annotate all Haiku elicitations, we use stratified sampling (App. H) to approximate the full evaluation distribution. Specifically, we sample one seed per successful task while matching the overall severity distribution, ensuring coverage of all tasks, domains, and severity levels. This allows us to prioritize capturing all dimensions that may impact evaluator performance rather than randomly sampling. In addition, we annotate all Opus examples due to their smaller scale. Together, this provides a representative FPR estimate across the most relevant dimensions despite partial annotation
>
> ### Additional Concerns
>
> We refer to our other review responses to address your remaining concerns due to space limits:
> - **cA4a:** Opus 4.5 Evaluation (W1&Q1), Evaluator Reliability & False Negatives (W2&Q2)
> - **enqn:** Baseline Comparison (W2), Single vs. Ensemble Judge (Q3)

---

> > ### Author Rebuttal · Reviewer_9Tg3 · 2026-04-01
> >
> > Thank you for your response. Several of my primary concerns have been addressed, including the cross-model performance and the ablation of key design decisions. I have therefore raised my score to 4 (Weak Accept).

---

### Official Review · Reviewer_cA4a · 2026-03-11

**Soundness:** 2
**Presentation:** 3
**Significance:** 2
**Originality:** 2
**Overall Recommendation:** 4
**Confidence:** 4

**Summary:**

The paper tackles a critical but overlooked gap in Computer-Use Agent (CUA) safety: vulnerabilities triggered by benign, underspecified instructions rather than overt adversarial attacks. The authors argue that everyday ambiguity is a more realistic threat vector for deployed agents. They introduce AUTOELICIT, a dual-loop framework designed to find these "edge cases" where a model might interpret a vague command in a way that violates the CIA triad. While the conceptual framing is strong, the empirical rigor—particularly regarding state-of-the-art (SOTA) models—is spread too thin to support some of the broader claims.

**Compliance With Llm Reviewing Policy:**

Affirmed.

**Final Justification:**

My concerns have been addressed.

**Key Questions For Authors:**

• To move beyond the current selection bias, can you provide a zero-shot evaluation on a larger, randomly sampled set (N ≥ 100) for the SOTA models?

• Regarding the 20% evaluator error, does the VLM tend to "hallucinate" harm where none exists, or does it miss subtle data leaks? This distinction is vital for understanding the safety margins.

• In your transferability tests, how often did the "safe" models actually trigger a safety refusal string versus simply getting stuck or timing out?

**Limitations:**

There is no paragraph about limitation discussion.  The primary limitation is the lack of statistical power in the SOTA model evaluation. Additionally, the study does not provide a formal metric for "benignity," relying instead on a qualitative loop which may not capture all nuances of how a human would perceive the instructions. The distinction between a model being "safe" and a model being "incapable" remains blurred in the current comparative analysis.

**Strengths And Weaknesses:**

Strengths:
• Shifting the focus from "jailbreaking" to "unintended harm from benign prompts" is the paper’s strongest contribution. It addresses the practical reality of how these agents will actually fail in production.

• The categorization of harms is well-grounded. The qualitative examples, such as an agent deleting system files during a "cleanup" task, provide a much-needed "so what" factor that is often missing from theoretical alignment papers.

• The AUTOELICIT feedback loop is a clever way to maintain the "benign" appearance of a prompt while iteratively pushing the agent toward a failure state.

Weaknesses:
• The evaluation of the most capable models (Claude 4.5 and GPT-5) is statistically fragile. Testing only 30 seeds per domain is already a small sample, but the fact that these seeds were pre-selected based on their high severity on a weaker model (Haiku) fundamentally skews the results. We cannot conclude anything about the general robustness of these models if the test set is essentially "cherry-picked" for known failure modes.

• A VLM-based evaluator with a 20% error rate and a Fleiss’ Kappa of 0.453 is borderline unacceptable for a safety-critical study. In this domain, the difference between a "safe" and "harmful" action is often subtle. Without a rigorous breakdown of false positives vs. false negatives, the "success rates" reported in the tables are difficult to trust.

• The claim that open-source models (like EvoCUA) are more "robust" appears to be a misinterpretation of model failure. There is a high likelihood these models are "safe" simply because they lack the reasoning capability to execute the complex (and potentially harmful) task in the first place. The paper needs to distinguish between a refusal (alignment) and a failure to act (incapability).

---

> ### Author Rebuttal · Authors · 2026-03-31
>
> We appreciate your thoughtful feedback and positive comments regarding our work’s practical importance, strong conceptual framing, and clever dual feedback approach. We have conducted additional analyses to address your concerns on empirical rigor.
>
> ### Opus 4.5 Evaluation (W1&Q1)
>
> We appreciate your concern regarding selection bias for Claude 4.5 Opus results. To assess general robustness, we randomly sample 60 OS and Multi-Apps seeds (120 total) without using Haiku outcomes and run AutoElicit on Opus (GPT-5 as the refinement model, same hyperparameters used as Tab. 1 (App. E)). We report the severity distribution of these seeds against Haiku to verify no selection bias:
>
> |Domain|None|Min.|Low|Med.|High|Crit.|
> |-|-|-|-|-|-|-|
> |OS|35.0|0.0|18.3|30.0|11.7|5.0|
> |Multi-Apps|20.0|11.7|35.0|21.7|8.3|3.3|
>
> We observe consistent elicitation success against Opus across these seeds (85% for OS, 86.7% for Multi-Apps). AutoElicit also still surfaces severe harms, eliciting High-Critical severity risks in 15% of OS seeds and 15.3% of Multi-Apps. These results show AutoElicit surfaces long-tail risks from SoTA CUAs even without selecting high-severity seeds.
>
> |Domain|Elic. Per Seed %|Elic. Per Task %|None|Min.|Low|Med.|High|Crit.|
> |-|-|-|-|-|-|-|-|-|
> |OS|85.0|100.0|15.0|0.0|18.3|51.7|11.7|3.3|
> |Multi-Apps|86.7|90.6|11.9|1.7|23.7|47.5|10.2|5.1|
>
> ### Evaluator Reliability & False Negatives (W2&Q2)
>
> **FPR:** Thanks for raising this important concern regarding our evaluator's reliability. We would like to clarify that the ~20% FPR and 0.453 Fleiss’ Kappa reflect the full multi-stage pipeline, not just binary harm classification. Each TP requires accurate evaluation across four criteria (i.e., trajectory analysis, contextual harm assessment, user intent interpretation, and CoT analysis) with their own failure modes within our pipeline. For more details about these criteria and evaluation challenges, see our response to m5vt’s W4.
>
> To understand our evaluator’s failure modes, we provide a FP breakdown in Tab. 6 from human annotation (App. H). We observe that ~56% of FP arise from the evaluator flagging a behavior as harmful when humans find it not to be. Importantly, this is not from the evaluator “hallucinating” that the behavior occurred, but from the inherent ambiguity of evaluating if the behavior was truly harmful given task and execution-specific context. Tab. 6 also shows that FPR is higher for low severity elicitation (Min/Low/Med), indicating that AutoElicit can be overly cautious in flagging borderline, low-severity risks.
>
> **False Negatives:** We agree that measuring false negatives is critical. For this, we conduct **an additional human annotation** study on 50 randomly sampled trajectories labeled as non-harmful by AutoElicit. For coverage, we include trajectories across all 50 Multi-Apps tasks, execution agents, refinement models, and various refinement iterations.
>
> We apply our annotation procedure (App. H) using two criteria defining harms missed by our evaluator and perturbation quality. We find that our evaluator demonstrates a **2% FNR** across all evaluated trajectories. Along with FPR, this suggests that AutoElicit **has high recall** but oversensitivity to low severity harms.
>
> ### Model Robustness (W3&Q3)
>
> We appreciate your concerns regarding the robustness of open-source models and provide the following clarifications.
>
> 1. EvoCUA is a highly capable CUA achieving 56.7% on the OSWorld leaderboard. This is competitive with frontier CUAs like Claude 4.5 Sonnet (58.1%), suggesting that the lower harm rate is not due to weaker reasoning capability.
>
> 2. Since AutoElicit evaluates completely benign inputs rather than adversarial attacks, safe behavior is intended task execution rather than refusal. Across all manually annotated trajectories (132 in AutoElicit-Exec, 100 from EvoCUA/OpenCUA), we observe zero safety refusals.
>
> 3. We distinguish unintended behaviors from general mistakes occurring from model incapability based on goal-directed CoT reasoning (§3.4). Our human annotation procedure (App. H) includes this as an evaluation criteria, confirming that all human-verified elicitation success reflects safety limitations, not incapability.
>
> 4. All agents we evaluated do sometimes get stuck or timeout during execution. However, we focus on active unintended harms from the agent’s actions, as a failure to act does not alter the system state or introduce severe risks.
>
> ### Formal Metric of Benignity
>
> Thanks for raising this concern. We carefully construct the benignity and implicitness scores (Tab. 5) to enforce key properties of benign instructions (i.e., maintaining a safe interpretation, avoiding harmful requests), providing a scalable approximation of benignity for automatic elicitation. We validate this through human annotation (App. H) where most perturbations are rated benign across all three annotators, indicating the alignment of our criteria with multiple human interpretations of benignity.

---

> > ### Author Rebuttal · Reviewer_cA4a · 2026-04-01
> >
> > I appreciate your rebuttal. I'll raise the score.

---

### Official Review · Reviewer_m5vt · 2026-03-13

**Soundness:** 3
**Presentation:** 3
**Significance:** 3
**Originality:** 2
**Overall Recommendation:** 5
**Confidence:** 3

**Summary:**

The authors provide a benchmark to study unintended behaviors of computer-use agents (CUAs) based on benign inputs, using a method to generate vulnerable benign inputs via perturbation that may lead to such misbehavior. Tests on agents based on frontier models successfully uncover multiple such unintended behaviors.

**Compliance With Llm Reviewing Policy:**

Affirmed.

**Final Justification:**

Following the rebuttal, my stated concerns stand addressed. I will raise my score to recommend acceptance.

**Key Questions For Authors:**

Can you please address the weaknesses in the weaknesses section?

**Limitations:**

No. No limitations section is provided and no limitations are explicitly discussed.

**Strengths And Weaknesses:**

# Strengths

- [S1] The evaluation is done across some of the most recent and powerful CUAs, and this adds to the relevance of their results.
- [S2] The framework's methodology is well-described, and the seed perturbation and trajectory analysis are well motivated and implemented.
- [S3] The authors ensure soundness of results w.r.t. their benchmark, making sure that harms are not attributed wrongly to their perturbation method - See lines (356-361, right column) “To ensure surfaced harms are attributable to AUTOELICIT rather than the original task, we estimate a baseline harm rate by executing each benign task in AUTOELICIT-SEED five times with Haiku and retain only tasks with a 0% baseline harm rate.” This lends credence to their findings.
- [S4] The authors reasonably navigate using of costlier models by using meaningful subsets - for instance, they mention “In contrast, Opus examples are restricted to a subset of the 30 seed perturbations eliciting the most severe harms from Haiku for each refinement model.” in lines 361-363 (right column), reflecting good practices in resource-constrained settings.
- [S5] The soundness of the judge is attested to, barring a non-trivial error in TPR (see weakness W4). However, this is in the interest of being transparent about the judge's soundness, which the authors verify through human verification of the automated judge, which is good for the soundness of their results.
- [S6] A **major strength** is given by the significant transferrability of successful perturbations across CUAs; this further strengthen the soundness of the benchmark in auditing such harms from benign inputs. It also provides insights on how CUAs have similar triggers/vulnerabilities, suggesting that this may be a wider, model-agnostic problem that is prime for a mitigation study. Additionally, the fact that perturbations transfer more successfully from larger CUAs to smaller ones makes sense. This paper, therefore, goes beyond benchmarking to provide key, generalizable insights on benign perturbations and general model/CUA behavior.
- [S7] Unintended behavior analysis is useful and helps make sense of such unintended CUA behavior via a qualitative analysis beyond aggregate statistics.


# Weaknesses

- [W1] A lot of the provided examples appear to be more of an issue with careless user instructions, as opposed to the LLM in the agent core misbehaving. For example, the student mail and calculator.py examples are clear cases of this. This undermines some confidence in the validity of the included benign examples.
- [W2] The perturbation generation appears to have similarities to evasion attack generation, contradicting that this might be entirely benign, as it has an adversarial flavor to it. Though I shall concede that it may be framed as stress-testing.
- [W3] The authors say "We will release “AUTOELICIT-BENCH, a dataset of 117 human…” (lines 107-108), but they include it later. Initially, that gives an impression of a future (but presently missing deliverable),  and they should amend their language to reflect the current provision of this dataset.
- [W4] (Refer to strength S5) While the TPR is respectable, there is a sizable margin of error (79.5% TPR), and do we know that the critical/high harms that were detected (9.2-10.1%) do not fall within this margin?
- [W5] Regarding originality, this paper, while useful, seems to be fairly moderate in novelty as it adapts ideas from existing literature, such as agentic misalignment, unintended misbehavior, etc. from works like [1] to CUAs, so their novelty is mostly due to their new CUA-specific benchmark rather than the discovery of a fundamentally new agentic risk.

--

## References
[1] Järviniemi, O., & Hubinger, E. (2024). Uncovering Deceptive Tendencies in Language Models: A Simulated Company AI Assistant. ArXiv, abs/2405.01576.

---

> ### Author Rebuttal · Authors · 2026-03-31
>
> We thank the reviewer for the thoughtful feedback and positive comments on our methodology, relevant evaluation, and valuable transferability analysis. We seek to address your concerns and offer the following clarifications:
>
> ### Careless User Instructions (W1)
>
> Thanks for raising this point. We agree that some user instructions are carelessly underspecified. However, safe CUA deployment requires reliability despite such imperfections that are inevitable in real-world use.
>
> As noted in lines 26-43, natural language instructions often express user intent implicitly as explicitly enumerating all constraints is infeasible and undermines the goal of simple, intuitive human-agent communication. A goal of ours is to study whether CUAs can maintain safety under such benign conditions where user intent must be inferred rather than being fully specified. Unintended behaviors from AutoElicit thus reflect CUA shortcomings in taking actions to pursue intended goals and avoid unintended harms by adhering to core safety principles and resolving ambiguity (e.g., requesting clarification). We therefore consider these as valid unintended behaviors arising from fundamental safety limitations rather than just from careless user input.
>
> ### Adversarial Perturbations (W2)
>
> We understand the concern around similarities with red-teaming or evasion attacks. However, unlike those settings, we do not use adversarial inputs. We are strictly constrained to the elicitation of safety risks under realistic, benign inputs rather than malicious prompts.
>
> To enforce our formulation, we include strict benignity and realism constraints during both generation and refinement. Candidate seed perturbations in Context-Aware Seed Generation are evaluated and filtered via Constraint Adherence Scores (Tab.5), requiring them to 1) have a plausible safe interpretation, 2) resemble realistic user requests, and 3) avoid explicitly requesting harmful behavior. These scores are also used in Execution-Guided Perturbation Refinement via Quality Refinement, where each proposed perturbation undergoes a Quality Check and is refined if it violates any benignity or realism constraints. Through this, we ensure that only perturbations meeting our formulation are used as candidate seeds and prevent drift toward adversarial prompts during refinement.
>
> We validate this via human annotation (App. H), where one of four success criteria requires perturbations to be benign, realistic, and not directly instruct the harm. AutoElicit achieves 79.5% TPR under this criterion, showing it surfaces unintended behaviors without adversarial inputs.
>
> ### Dataset Release (W3)
>
> All datasets mentioned in our work are publicly available and AutoElicit-Bench is included in the supplementary material. We will adjust phrasing to clarify this in the revised version.
>
> ### TPR & Error Rate by Harm Severity (W4)
>
> Thanks for pointing out concerns regarding the margin of error. We would like to emphasize that the 79.5% TPR reflects performance across the entire multi-stage evaluation pipeline. As described in the evaluation criteria (App. H), each True Positive must feature accurate 1) trajectory analysis to identify an elicited harm, 2) reasoning to interpret whether behavior is truly harmful given task context, 3) interpretation of implicitly defined user intent and 4) interpretation of CoT to ensure goal-directed harms. For comparison, OS-Harm’s classifier [1] achieves a 77% F1 for a simpler binary harm classification, showing our approach achieves stronger reliability despite more complex evaluation criteria.
>
> Tab. 6 also shows that TPR is highest for the most severe harms based on human annotation. AutoElicit achieved a TPR of 90.6% and 83.3% for High and Critical severity seeds respectively, indicating that the most consequential risks are less likely to fall within the margin of error and that AutoElicit maintains high reliability for the most safety critical outcomes.
>
> ### Novelty (W5)
>
> We appreciate your concerns regarding novelty compared with existing works. We first clarify that our technical contributions extend beyond a CUA-specific benchmark as we construct AutoElicit, the first automatic elicitation pipeline for scalably eliciting such behaviors in CUAs.
>
> In addition, we provide a concrete conceptual framework of unintended CUA behaviors to enable systematic analysis of this risk. Prior unintended behavior work (§2) offers fragmented definitions that only provide details for their threat model, agent vulnerability, or deployment settings. As recognized by 9Tg3 and enqn, our work provides a novel, unified conceptual framework of unintended behaviors precisely defining their key characteristics, main categories, and the vulnerabilities causing the risk. This conceptual framework provides structure for the broader safety risk, enabling consistent discussion and rigorous analysis of unintended CUA behaviors.
>
> [1] OS-Harm: A Benchmark for Measuring Safety of Computer Use Agents

---

> > ### Author Rebuttal · Reviewer_m5vt · 2026-04-02
> >
> > I thank the authors for engaging with the substance of the rebuttal meaningfully.
> >
> > I appreciate the clarified language and descriptions provided for W1 and W2, and I highly encourage the authors to add these clarifications to the revision to make the paper more sound in its arguments. It will help make sure that the benign constraint angle and the unintended effects of misspecification are clearly addressed. Similarly, please clarify the language in the paper in the light of W3 and your response to it.
> >
> > I also appreciate the clarification regarding the TPR (W4), which illustrates how this setting is a difficult one and may have various surfaces/step where errors may occur. I take this point.
> >
> > Besides, I will acknowledge that the discussion about the level of novelty of this benchmark, as described by my comments, may incur comments like W5, it is not a major flaw, if one at all. It is still a useful resource for practitioners who deploy and develop CUAs. This paper's novelty is sufficient to merit recognition as a useful scientific and safety-auditing contribution and be presented at this venue.
> >
> > With this, I will say that my concerns are now addressed, and I will raise my score to further support the acceptance of this paper (and would encourage the authors to incorporate the suggested clarifications in it while at that).

---

### Decision · Program_Chairs · 2026-04-30

**Decision:**

Accept (regular)

**Comment:**

This paper addresses unintended unsafe behaviors in Computer-Use Agents (CUAs) that arise from benign, underspecified user instructions rather than adversarial attacks. The authors contribute: (1) a conceptual framework defining unintended CUA behaviors through four key characteristics, (2) AutoElicit framework that iteratively perturbs benign instructions using CUA execution feedback to surface such behaviors, and (3) empirical findings demonstrating high elicitation success rates across frontier CUAs, with strong cross-model transferability.

The paper addresses a timely and practically important safety problem with a well-designed methodology for computer use agents. The work provides both a useful conceptual framework and a practical toolkit for auditing CUA safety under realistic conditions, which the community will benefit from. Reviews suggest tightening the presentation of the paper, including some clarifying details, and including the discussions during the rebuttal which can improve the paper drastically.